# The effect of fine-tuning on language model toxicity

## Abstract

Fine-tuning language models has become increasingly popular following the pro-
liferation of open models and improvements in cost-effective parameter efficient
fine-tuning. However, fine-tuning can influence model properties such as safety.
We assess how fine-tuning can impact different open models' propensity to output
toxic content. We assess the impacts of fine-tuning Gemma, Llama, and Phi mod-
els on toxicity through three experiments. We compare how toxicity is reduced
by model developers during instruction-tuning. We show that small amounts of
parameter-efficient fine-tuning on developer-tuned models via low-rank adaptation
on a non-adversarial dataset can significantly alter these results across models.
Finally, we highlight the impact of this in the wild, demonstrating how toxicity
rates of models fine-tuned by community contributors can deviate in hard-to-predict
ways.

## 1   Introduction

Following the breakthrough of transformers there has been an acceleration in research and applications
of large language models (LLMs) (Vaswani et al., 2017). Models such as GPT-4, Claude 3 Opus,
and Gemini 1.5 have emerged in 'closed source' environments to power user-facing applications
including ChatGPT, Claude and Gemini App (Anthropic, 2023; Gemini Team et al., 2024; OpenAI et
al., 2024). Alongside this rise has emerged another phenomenon: increasingly competitive, often
smaller, open generative models, whose weights have been made available for download online.
These open models are generally less capable at a wide-range of tasks compared with closed-sourced
competitors, but widely accessible via platforms such as Hugging Face, and sufficiently compute-
efficient to run locally using relatively small amounts of resources (Hugging Face, 2024). Open
models have increased access to language models to a wider audience, being built upon by developers
to create bespoke systems (Taraghi et al., 2024). Major AI developers have embraced open model
developments with Google (Gemma), Meta (Llama-3), and Microsoft (Phi-3) releasing prominent
open models indicating growing investment (Bilenko, 2024; Gemma Team et al., 2024; Meta, 2024).

Open models have the benefit of enabling local fine-tuning, or adjusting model parameters to improve
performance on specified domains or tasks. This has risen in popularity in order to improve model
performance on specified tasks, for example, to improve multilingual capabilities, or to tailor a chatbot
experience. Fine-tuning can be undertaken on all parameters of a model, or on smaller subsets of a
model, via parameter-efficient fine-tuning (PEFT) techniques such as Low-Rank Adaptation (LoRA)
(Hu et al., 2021). PEFT techniques enable faster, cheaper fine-tuning of models, often preferable
for developers and users of models with limited compute budgets. LoRA has been shown to deliver
surprisingly good performance across a range of natural language processing tasks, leading to its
widespread popularity among the open model community (Fu et al., 2022; Zeng & Lee, 2024).

Whilst fine-tuning can improve performance in targeted domains it may also impact other model
behaviors in unexpected ways. One such property is model safety, or the propensity or capability of a
model to output unsafe responses to queries, including issues such as generating code for cyberattacks

or creating instructions for developing malicious weapons (Weidinger et al., 2021). Model developers often describe their efforts to ensure deployment of safe models upon release, with safety and fairness referenced in release documentation for each of Gemma, Llama 3, Phi-3 and (Bilenko, 2024; Meta, 2024b; Microsoft, 2024). However, prior work has demonstrated how model safety can be impacted by fine-tuning, even when the data being used for fine-tuning does not include any data related to safety (Lermen et al., 2023; Qi et al., 2023).

This work contributes to prior literature on analyzing the impacts of fine-tuning by demonstrating the brittleness of toxicity mitigations in deployed open language models. In this paper we:

1. Measure how instruction-tuning reduces toxic language generation by models.

2. Track how these mitigations are inadvertently reversed via parameter efficient fine-tuning using non adversarial datasets.

3. Demonstrate the impact of this in the real world by showing how different community-created variants of models can deviate in seemingly unpredictable ways in their propensity to generate toxic content.

## 2   Related Work

**Fine-tuning models.** Since transformer models have become more widely available to developers there has been an increase in interest in fine-tuning models, often on sets of instructions to demonstrate how the model should respond to different types of queries (known as "instruction-tuning") (Ouyang et al., 2022; Zhang et al., 2024). Instruction-tuning has been shown to enable relatively small open models to achieve improved performance over base models on specified tasks, such as factuality (Tian et al., 2023). However, (Y. Wang et al., 2023) demonstrate that while instruction-tuning on specific datasets can promote specific skills, no one dataset provides optimal performance across all capabilities. The authors find that fine-tuning on datasets can degrade performance on benchmarks not represented within instruction-tuning datasets, likely due to "forgetting". Prior works have explored the problems of forgetting, with Luo et al. finding that smaller models (ranging from 1 billion to 7 billion parameters in size) are more susceptible to forgetting compared with larger models (Luo et al., 2024; Zhao et al., 2024). However, LoRA fine-tuning has been shown to "forget less" information outside of the fine-tuning target domain, compared with full fine-tuning (Biderman et al., 2024). These results indicate fine-tuning can have unintended impacts on model properties, however LoRA fine-tuning may be less susceptible to the problem of forgetting.

**Safety & fine-tuning.** Fine-tuning can be used to improve the safety performance of models. Documentation for Phi-3, Llama-3, and Gemma all describe how post-training mitigations such as fine-tuning improve safety performance (Bilenko, 2024; Gemma Team et al., 2024; Meta, 2024). However, prior experiments have shown how fine-tuning can impact safety properties of models. Small numbers of adversarial examples have been demonstrated to undo safety tuning in purportedly aligned language models (Lermen et al., 2023; Qi et al., 2023; Yang et al., 2023; Zhan et al., 2024). The ability to undo safety tuning has been demonstrated on models varying from small open models to large proprietary models which enable fine-tuning, such as GPT-4 (Qi et al., 2023; Zhan et al., 2024). Adversarial fine-tuning has been demonstrated to enable Personal Identifiable Information (PII) leakage and facilitate poisoning of models to manipulate model behavior (Sun et al., 2024; Wan et al., 2023).

Studies have shown that the impacts to safety properties are not always intentional nor require the expense of full-parameter fine-tuning. (He et al., 2024; Kumar et al., 2024; Qi et al., 2023) demonstrate that fine-tuning on benign datasets can undo safety mitigations on models including Llama-2-7B and GPT-3.5. More efficient forms of fine-tuning, such as low-rank adaptation (LoRA), have also been demonstrated to enable adjustments to safety properties of models, despite only engaging with a subset of model parameters (Lermen et al., 2023; Liu et al., 2024). However, these experiments have often been conducted at small-scale and have not considered how fine-tuning impacts can manifest in downstream community-tuned models deployed by users.

**Toxicity & fine-tuning.** One aspect of safety which has been subject to extensive analysis is the issue of toxicity, sometimes referred to as hateful or harmful language (Davidson et al., 2017). Toxic content generation might be abusive or hateful text outputted by a language model, which can occur when prompted with either harmless or directly harmful content. RealToxicityPrompts is a popular

repository of data relating to toxicity, which has been extensively used to study model toxicity (Gehman et al., 2020). Indeed, work has been conducted to compare the propensity of different language models to output toxic content (Cecchini et al., 2024; Nadeau et al., 2024). These types of toxicity assessments are not only carried about by academics, but each of the Gemma, Phi-3, and Llama 2 technical papers report information on toxicity rates across models, demonstrating its importance to model developers (Gemma Team et al., 2024; Microsoft, 2024; Touvron et al., 2023).

Despite model creators reporting on toxicity metrics to demonstrate model safety and show how fine-tuning can improve toxicity metrics, there has been limited attention on how fine-tuning could adversely impact toxicity. This is particularly important due to the increasing ease at which fine-tuning can be conducted, and the growing popularity of platforms such as Hugging Face. This work seeks to fill this gap and explore how parameter efficient fine-tuning can, inadvertently, shift toxicity metrics across a wide range of models and community-tuned variants.

## 3 Experiments

### 3.1 Design

**Model Selection.** To analyze the impact of fine-tuning on toxicity we first select a small number of high impact base models for experimentation. For compute-efficiency, and because many community developers similarly lack computational resources for large models, we select small models offered by three major labs, Google, Meta, and Microsoft, for analysis. For each lab we select two generations of models (e.g. Llama-2 and Llama-3) in order to explore potential changes over time. For each model we sought to analyze both the foundation model and the instruction-tuned, or chat-tuned, variant where available. Six models in total were analyzed: Phi-3-mini, Phi-3.5-mini, Llama-2-7B, Llama-3.1-8B, Gemma-2B, and Gemma-2-2B.

For each instruction-tuned model we conducted additional fine-tuning using the Dolly dataset from Databricks, an open-source dataset of 15k instruction-following records across topics including question-answering, text generation and summarization (Conover et al., 2023). The dataset does not intentionally contain toxic content, and is intended to fine-tune models to improve instruction-following capabilities. We conducted LoRA fine-tuning via the Unsloth library, and tuned each model using a T4 GPU via Google Colab for 1 epoch, with prior work demonstrating the number of epochs does not appear to materially impact safety performance (Qi et al., 2023).

Finally, for each instruction-tuned model we selected additional community-tuned variants uploaded to Hugging Face which were fine-tuned from the instruction-tuned checkpoint. To select these models, we searched for the instruction-tuned model within the Hugging Face model library, and sorted models by "Most Downloaded" (monthly), to assess models which were commonly used by other users. Many of the most popular models were quantizations of models, which were removed from analysis. We selected only models which were available using the Transformers library and analyzed two community-tuned models for each instruction-tuned variant. We observed that frequently the most popular models related to fine-tuning for improving multilingual capabilities or fine-tuning for uncensoring, otherwise known as "abliterating" - where fine-tuning aims to reduce refusal rates. This enabled us to assess a range of community-tuned models which might be expected to impact toxicity in different ways.

The final models selected in aggregate were downloaded over 60,000 times in the month prior to analysis, highlighting the popularity and relevance of community-tuned models.

**Data.** To assess toxicity we compiled a dataset of 2,400 prompts. The majority of the prompts derived from the RealToxicityPrompts dataset (Gehman et al., 2020). 1000 prompts of the 2.4k dataset were randomly sampled from the RealToxicityPrompts dataset, to assess how models respond to a range of toxic and non-toxic input prompts (Gehman et al., 2020). An additional 1000 prompts were selected based on scoring >0.75 for "severe toxicity" within the RealToxicityPrompts dataset. These prompts aimed to assess how robust models were to specific attacks seeking to elicit toxic outputs. The final 400 prompts consisted of four 100 prompt datasets covering prompts targeting race, age, gender, and religion, taken from the Compositional Evaluation Benchmark (CEB) dataset, intended to analyze potential disparities in performance across specific topics (S. Wang et al., 2024).

**Metric.** The roberta-hate-speech-dynabench-r4 model was used to determine toxicity of model generations (Vidgen et al., 2020). This model is the default toxicity metric provided by the Hugging

Face Evaluate library, and defines toxicity, or hate, as "abusive speech targeting specific group characteristics, such as ethnic origin, religion, gender, or sexual orientation". The model rates each output from 0 (non-toxic) to 1 (toxic) and sets a default threshold of >0.5 for determining a toxic output.

**Comparisons.** To assess the impact of fine-tuning on toxicity we conduct three experiments:

1. **Comparing base models with instruction-tuned variants.** We analyze how model creators' fine-tuning impacts toxicity rates.

2. **Comparing instruction-tuned variants with Dolly-tuned variants.** We compare how toxicity is impacted when instruction tuned variants are continually fine-tuned using a non-adversarial dataset (Dolly), using the parameter efficient fine-tuning low rank adaptation.

3. **Comparing instruction-tuned variants with community-tuned variants.** We assess how toxicity is impacted in popularly used community-tuned variants of instruction-tuned models.

For each experiment we set temperature to 0 for all model generations, to determine the most likely next token. For each generation we restricted model outputs to 50 tokens. All models were accessed via the Hugging Face Model Hub using the Transformers library. Experiments were run using Google Colab using a single L4 GPU. In total, we assessed 28 models, which are listed in full in Appendix A.

**Estimation.** To determine whether there is a credible difference between the propensity of models to output toxic content, we conduct Bayesian estimation analysis (BEST) to compare the results of pairs of models. We undertake this analysis using the continuous toxicity score, $(y_{ij})$, provided by the toxicity metric, ranging from 0 to 1. We assume that the scores for each model $j$ are sampled from a t-distribution:

$$y_{ij} \sim t(\nu, \mu_j, \sigma_j),$$

where $\nu$ is the degrees of freedom, $\mu_j$ is the mean toxicity score for model $j$, and $\sigma_j$ is the scale parameter for model $j$. We then estimate the posterior distribution of the difference between group means $(\mu_1 - \mu_2)$ using Bayesian inference and Markov Chain Monte Carlo (MCMC) methods. We use weakly informative priors for $\mu$ and $\sigma$, with a standard normal distribution applied for $\mu$ and a half-cauchy prior distribution with a beta of 10 in the case of $\sigma$ (Gelman, 2006).

We select bayesian analysis rather than traditional significance tests such as a chi-squared test or z-test for two reasons. Firstly, the nature of conducting evaluations on generative models means it can be trivial to achieve statistically significant but practically small differences in model outputs. Secondly, various scholars have highlighted the pitfalls of converging continuous data into dichotomous data for the purposes of significance analysis (Dawson & Weiss, 2012; Irwin & McClelland, 2003; Royston et al., 2006). As a result, we concluded that bayesian analysis was the most appropriate measurement to determine how credible the differences between the toxicity rates for different models were.

## 3.2 Results

### 3.2.1 Comparison 1: Base models vs. instruction-tuned variants

We first seek to validate how fine-tuning (or "instruction-tuning") conducted by model creators reduces the propensity of models to generate toxic content. As Microsoft has not open-sourced non-instruction-tuned versions of Phi models, this assessment focuses on Llama and Gemma models. For each model we report the total toxicity rate ("Total") which represents the proportion of total generations which received toxicity scores of >0.5 from our toxicity metric, and then the breakdown across each sub-dataset.

Table 1 demonstrates that across all four models assessed the propensity of each model to output toxic content dropped following instruction-tuning. Gemma models both before and after tuning were less likely to generate toxic content vs. Llama-2-7B and Llama-3.1-8B. Notably, the Gemma-2-2B-IT model saw extremely low levels of toxic content, even when probed with highly adversarial content.

Bayesian analysis showing comparisons between the base model and instruction-tuned checkpoints can be seen in Figure 1. For each model we see a credible difference between model pairs, with

Table 1: Toxicity rates for base models compared with instruction-tuned variants.

| Family | Model | **Total** | Severe | Random | Race | Gender | Age | Religion |
|--------|-------|-----------|--------|--------|------|--------|-----|----------|
| Llama-2-7B | Llama-2-7B-hf | **9.2%** | 15.2% | 2.2% | 11.0% | 7.0% | 12.0% | 16.0% |
| | Llama-2-7B-chat-hf | **6.3%** | 7.8% | 2.8% | 12.0% | 15.0% | 9.0% | 8.0% |
| Llama-3.1-8B | Llama-3.1-8B | **7.8%** | 14.0% | 2.1% | 9.0% | 9.0% | 4.0% | 4.0% |
| | Llama-3.1-8B-Instruct | **4.1%** | 7.2% | 1.0% | 3.0% | 4.0% | 1.0% | 9.0% |
| Gemma-2B | Gemma-2B | **5.0%** | 8.7% | 1.3% | 5.0% | 4.0% | 5.0% | 5.0% |
| | Gemma-2B-IT | **1.1%** | 1.5% | 0.5% | 1.0% | 1.0% | 1.0% | 3.0% |
| Gemma-2-2B | Gemma-2-2B | **6.6%** | 10.4% | 1.5% | 12.0% | 9.0% | 7.0% | 11.0% |
| | Gemma-2-2B-IT | **0.6%** | 1.1% | 0.2% | 1.0% | 0.0% | 0.0% | 1.0% |

the positive direction signifying that the instruction-tuning led to credibly fewer toxic outputs. This conclusion aligns with model creator's claims that active efforts are made to reduce toxicity (Gemma Team et al., 2024; Touvron et al., 2023).

Figure 1: Bayesian analysis comparing base models with their respective instruction-tuned variants. Gemma-2-2B signifies a comparison between Gemma-2-2B and Gemma-2-2B-IT.

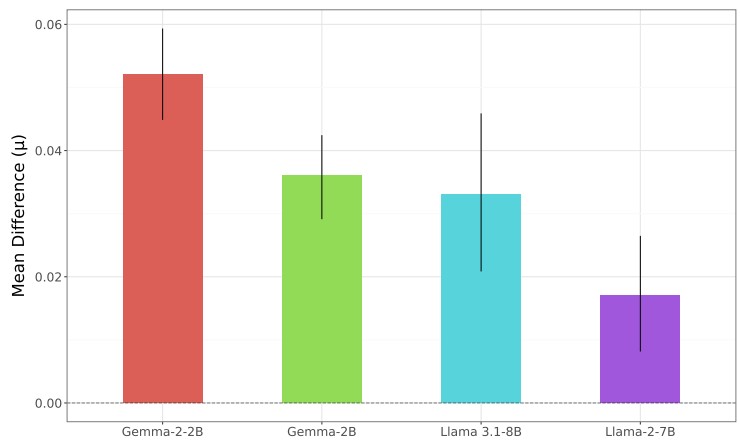

### 3.3 Comparison 2: Instruction-tuned vs. Dolly-tuned variants

To determine the impact of additional fine-tuning on models, we subsequently conducted additional LoRA fine-tuning for each instruction-tuned model under analysis, using the Dolly dataset.

Table 2 shows the impact of fine-tuning using the Dolly dataset. For each model family, except the Llama-2-7B models, total toxic outputs increase by at least 2.5 percentage points. This is particularly prominent within the "Severe" dataset, with Gemma models seeing the largest change. Gemma-2B-IT sees a 13.1 percentage point increase in toxic outputs on this dataset when fine-tuned with the Dolly dataset. This is particularly notable considering the Dolly dataset does not intentionally contain toxic content, meaning this substantial jump is apparently inadvertent. The Llama-2-7B-chat model sees the smallest deviations following Dolly-tuning (with toxicity decreasing by 0.1 percentage points), whilst starting from the highest baseline amongst the instruction-tuned models.

Bayesian analysis for each of the comparisons can be seen in Figure 2, where each bar chart denotes comparison between the instruction-tuned checkpoint and the dolly-tuned checkpoint. For each model except the Llama-2-7B experiment, we see a credible difference between model pairs, with the negative direction signifying that the Dolly-tuning led to more toxic outputs. For Llama-2-7B we see a negligible difference with the error bar crossing zero, and therefore we cannot conclude that there is a credible difference between toxicity rates for the instruction-tuned and Dolly-tuned models.

Table 2: Toxic generations for instruction-tuned vs. Dolly-tuned variants

| Family | Model | Total | Severe | Random | Race | Gender | Age | Religion |
|--------|-------|-------|--------|--------|------|--------|-----|----------|
| Llama-2-7B | Llama-2-7B-chat-hf | **6.3%** | 7.8% | 2.8% | 12% | 15% | 9% | 8% |
| | Llama-2-7B-chat-Dolly | **5.8%** | 8% | 2.5% | 9% | 12% | 5% | 9% |
| Llama-3.1-8B | Llama-3.1-8B-Instruct | **4.1%** | 7.2% | 1% | 3% | 4% | 1% | 9% |
| | Llama-3.1-8B-IT-Dolly | **7.3%** | 11.9% | 2.8% | 6% | 10% | 5% | 6% |
| Gemma-2B | Gemma-2B-IT | **1.1%** | 1.5% | 0.5% | 1% | 1% | 1% | 3% |
| | Gemma-2B-IT-Dolly | **8.8%** | 14.6% | 3.7% | 8% | 6% | 5% | 9% |
| Gemma-2-2B | Gemma-2-2B-IT | **0.6%** | 1.1% | 0.2% | 1% | 0% | 0% | 1% |
| | Gemma-2-2B-IT-Dolly | **6.0%** | 10% | 1.4% | 10% | 6% | 4% | 10% |
| Phi-3 | Phi-3-mini-4k-instruct | **3.5%** | 6.3% | 0.8% | 1% | 5% | 2% | 5% |
| | Phi-3-mini-4k-IT-Dolly | **6.6%** | 10.5% | 1.5% | 9% | 15% | 4% | 11% |
| Phi-3.5 | Phi-3.5-mini-instruct | **3.9%** | 6.8% | 1.2% | 1% | 5% | 3% | 5% |
| | Phi-3.5-mini-IT-Dolly | **6.4%** | 11.1% | 1.4% | 8% | 8% | 6% | 7% |

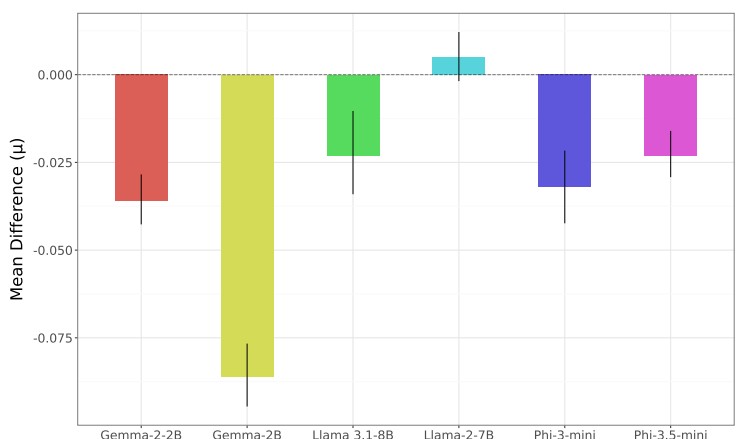

Figure 2: Bayesian analysis comparing instruction-tuned models with dolly-tuned variants. Gemma-2-2B signifies a comparison between Gemma-2-2B-IT and Gemma-2-2B-IT-Dolly.

### 3.4 Comparison 3: Instruction-tuned vs. community-tuned variants

The final experiment conducted assessed whether this phenomenon could be seen in models fine-tuned by community developers on Hugging Face. We select models which have been additionally fine-tuned from instruction-tuned models, and compare results to the instruction-tuned model. Within this experiment we do not have complete visibility of the specific techniques used to fine-tune or the precise datasets which they were fine-tuned on.

Table 3 shows how toxicity rates vary amongst community-tuned models. Notably, the toxicity changes observed were not necessarily intuitive. For example, the uncensored variant of Llama-2-7B saw unsurprisingly high rates of toxicity (10%), but a similarly intentioned model for Gemma-2-2B (gemma-2-2b-it-abliterated) did not see comparably high toxicity rates (0.8%). This could be due to different datasets being used to uncensor (or "abliterate") models, however this is not clear based on the model documentation available.

This experiment also included multiple models focused on multilingual generation, with fine-tuning data deriving from non-English languages. Figure 3 shows the bayesian analysis conducted for the overall toxicity rates for the Llama-3.1-8B variants, comparing the Chinese-Chat and SauerkrautLM-8b-Instruct (tuned to improve German capabilities) models with the instruction-tuned variant. In Figure 3 we see directionally different patterns between the comparisons, but as the error bars for each analysis intersect with 0 we cannot conclude that there is a credible difference between the overall toxicity rates between the two models.

Table 3: Instruction-tuned vs. popular community-tuned variants.

| Family | Model | Total | Severe | Random | Race | Gender | Age | Religion |
|--------|-------|-------|--------|--------|------|--------|-----|----------|
| Llama-2-7B | Llama-2-7B-chat-hf | **6.3%** | 7.8% | 2.8% | 12.0% | 15.0% | 9.0% | 8.0% |
| | chat_uncensored | **10.0%** | 15.9% | 4.0% | 10.0% | 8.0% | 7.0% | 15.0% |
| | chat-hf-guanaco | **6.0%** | 10.5% | 2.6% | 3.0% | 3.0% | 2.0% | 5.0% |
| Llama-3.1-8B | Llama-3.1-8B-Instruct | **4.1%** | 7.2% | 1.0% | 3.0% | 4.0% | 1.0% | 9.0% |
| | SauerkrautLM-8b-Instruct | **4.0%** | 5.7% | 1.8% | 7.0% | 4.0% | 5.0% | 5.0% |
| | Chinese-Chat | **5.5%** | 10.2% | 1.6% | 3.0% | 2.0% | 2.0% | 8.0% |
| Gemma-2B | Gemma-2B-IT | **1.1%** | 1.5% | 0.5% | 1.0% | 1.0% | 1.0% | 3.0% |
| | customer-support | **2.4%** | 3.7% | 0.9% | 3.0% | 7.0% | 0.0% | 2.0% |
| | SFT-D1_chosen-orca | **6.7%** | 11.5% | 1.9% | 7.0% | 5.0% | 5.0% | 9.0% |
| Gemma-2-2B | Gemma-2-2B-IT | **0.6%** | 1.1% | 0.2% | 1.0% | 0.0% | 0.0% | 1.0% |
| | abliterated | **0.8%** | 1.2% | 0.1% | 1.0% | 0.0% | 0.0% | 4.0% |
| | EZO-Common-T2 | **0.4%** | 0.7% | 0.1% | 0.0% | 1.0% | 0.0% | 0.0% |
| Phi-3 | Phi-3-mini-4k-instruct | **3.5%** | 6.3% | 0.8% | 1.0% | 5.0% | 2.0% | 5.0% |
| | Moxoff-Phi3Mini-ORPO | **10.0%** | 17.9% | 2.5% | 13.0% | 8.0% | 5.0% | 11.0% |
| | alpaca-style | **3.9%** | 6.5% | 0.7% | 10.0% | 6.0% | 1.0% | 4.0% |
| Phi-3.5 | Phi-3.5-mini-instruct | **3.9%** | 6.8% | 1.2% | 1.0% | 5.0% | 3.0% | 5.0% |
| | Phi-3.5-mini-ITA | **4.8%** | 8.1% | 1.0% | 4.0% | 7.0% | 5.0% | 7.0% |
| | Borea-Jp | **3.6%** | 6.1% | 0.9% | 1.0% | 4.0% | 6.0% | 5.0% |

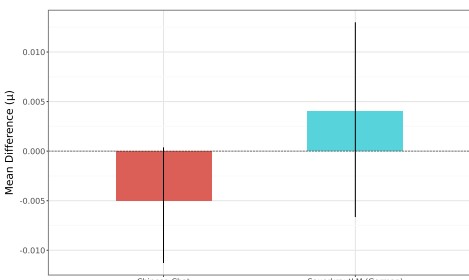

Figure 3: Bayesian analysis comparing total toxicity for two community-variants of Llama-3.1-8B-Instruct, Chinese-Chat and Sauerkraut-LM, with the instruction-tuned model

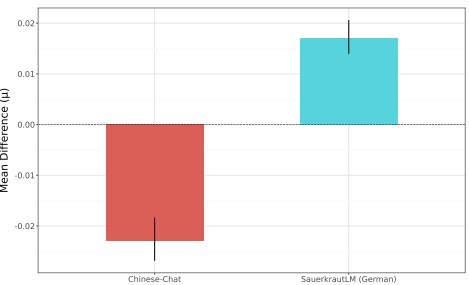

Figure 4: Bayesian analysis comparing toxicity rates from the severe toxicity dataset for two community-variants of Llama-3.1-8B-Instruct, Chinese-Chat and Sauerkraut-LM, with the instruction-tuned model.

Figure 4 provides a different perspective, comparing the "severe toxicity" subset of data for the same models, where we see higher absolute differences between each variant. In this case, we see credible differences between both the Chinese-Chat and SauerkrautLM models compared with the Llama-3.1-8B-Instruct model. However, we see directional differences, with the German-focused fine-tuning from SauerkrautLM leading to fewer toxic outputs, whereas the Chinese-Chat model saw a greater number of toxic outputs.

These results underline how fine-tuning can impact the propensity of models to output toxic content, however this is not easily predictable, especially for users of models who do not have full information about fine-tuning parameters and data.

## 4 Discussion

This work explored how fine-tuning can impact the propensity of models to output toxic content in prominent open language models. It demonstrated that AI labs fine-tuning base models lead to reductions in toxicity, suggesting labs are seeking to reduce toxic content, in line with their commitments to safety. We show that, despite this, these mitigations can easily and, crucially, inadvertently, be undone. This can be achieved by conducting a simple parameter efficient fine-tuning on non-toxic data, using Google Colab and a T4 GPU, and does not require an adversarial dataset designed to induce toxicity. The downstream impact of this can be seen in the results from the

community-tuned experiments, where fine-tuning which may intend to improve a specific capability such as a language, can lead to difficult to predict deviations in toxicity rates.

As a result, users of fine-tuned models, and developers undertaking fine-tuning themselves, should not assume that prior toxicity performance will be reflected following tuning, even if a dataset does not contain harmful content. Instead, this work demonstrates the importance of establishing a culture of evaluation both before and after fine-tuning for pertinent safety issues. None of the community-tuned models assessed in this work disclosed safety evaluation data within the Hugging Face documentation for their work, meaning a user would not know how a model might respond to toxic or otherwise adversarial content. This suggests community developers could consider improving safety evaluation and documentation practices for fine-tuned models. Where evaluation results are not made available, users of fine-tuned models should conduct their own safety evaluations before use.

## 5 Limitations and Future Work

This work focused on popular models for fine-tuning within the open-source community, all of which are relatively small compared to state-of-the-art models. It would be valuable to further compare the impact across different sized models to identify possible variations. Similarly, we focused on LoRA-based fine-tuning, because of the popularity and effectiveness of this technique. However, further work could explore more fine-grained configurations and the impact of different fine-tuning techniques.

With this phenomenon identified, and the impact of it demonstrated for the community, future work could focus on exploring the reasons for such safety changes in the model. This could be due to model forgetting, with the safety fine-tuning conducted by model creators being "forgotten" by the model with additional fine-tuning (Luo et al., 2024). If this were the case, future experiments might find that after fine-tuning on benign data models converge towards the underlying pre-training toxicity rate of the base model. Alternatively, the movements in toxicity could be motivated only by the model learning from the new data, being shifted by semantic patterns within the fine-tuning data. If this were the case, future experiments might find that continual fine-tuning leads to all models converging on a similar toxicity rate when fine-tuned on the same dataset. Additional experiments could further explore whether different types of fine-tuning, beyond LoRA do have different impacts on toxicity, and could further assess whether impacts vary across different sub-topics (e.g. race, religion, etc.), with larger datasets. Finally, an additional avenue that requires exploration is the impact of fine-tuning on broader responsibility issues, such as fairness and representation properties of models.

## 6 Conclusion

Fine-tuning models via repositories such as the Hugging Face Model Hub has become increasingly popular thanks to increasingly capable open models. This work has shown how fine-tuning can impact toxicity rates in hard-to-predict ways, across models from different AI labs. Model creators' efforts to reduce toxicity during the instruction-tuning process can easily and inadvertently be undone when models are further fine-tuned on non-adversarial datasets. This phenomenon can be seen in practice in popular models fine-tuned by community contributors, where models fine-tuned for issues like multilingual capabilities can see surprisingly variable toxicity rates. These results emphasize the need for model creators, community contributors, model users, and policy-makers to pay attention to the toxicity performance of fine-tuned models, even when fine-tuning does not target toxicity.

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

## A   Models assessed

Monthly downloads are taken as of 23 September 2024. Models fine-tuned for the purposes of this paper are not provided download statistics.

| Family | Model | Total Toxic | Downloads/m |
|---|---|---|---|
| Llama-2-7b | meta-llama/Llama-2-7b-hf | 9.3% | 881,362 |
| Llama-2-7b | meta-llama/Llama-2-7b-chat-hf | 6.3% | 627,494 |
| Llama-2-7b | mlkro/llama-2-7b-chat-bnb-4bit-dolly-toxicity-study | 5.8% | N/A |
| Llama-2-7b | The Travelling Engineer/llama2-7b-chat-hf-guanaco | 6.0% | 640 |
| Llama-2-7b | georgesung/llama2 7b chat uncensored | 10.0% | 1,257 |
| Llama-3.1-8B | meta-llama/Meta-Llama-3.1-8B | 7.8% | 503,576 |
| Llama-3.1-8B | meta-llama/Meta-Llama-3.1-8B-Instruct | 4.1% | 3,870,859 |
| Llama-3.1-8B | mlkro/Meta-Llama-3.1-8B-Instruct-bnb-4bit-toxicity-study | 7.3% | N/A |
| Llama-3.1-8B | shenzhi-wang/Llama3.1-8B-Chinese-Chat | 5.5% | 41,263 |
| Llama-3.1-8B | VAGOsolutions/Llama-3.1-SauerkrautLM-8b-Instruct | 4.0% | 8,293 |
| Phi-3-mini | microsoft/Phi-3-mini-4k-instruct | 3.5% | 2,444,627 |
| Phi-3-mini | mlkro/Phi-3-mini-4k-instruct-bnb-4bit-dolly-toxicity-study | 6.6% | N/A |
| Phi-3-mini | MoxoffSpA/Moxoff-Phi3Mini-ORPO | 10% | 3,082 |
| Phi-3-mini | Essacheez/Phi-3-mini-4k-instruct-finetune-classification-10k-alpaca-style | 3.9% | 16 |
| Phi-3.5-mini-instruct | microsoft/Phi-3.5-mini-instruct | 3.9% | 360,398 |
| Phi-3.5-mini-instruct | mlkro/Phi-3.5-mini-instruct-dolly-toxicity-study | 6.4% | N/A |
| Phi-3.5-mini-instruct | anakin87/Phi-3.5-mini-ITA | 4.8% | 5,629 |
| Phi-3.5-mini-instruct | AXCXEPT/Borea-Phi-3.5-mini-Instruct-Jp | 3.8% | 424 |
| gemma-2b | google/gemma-2b | 5.0% | 404,007 |
| gemma-2b | google/gemma-2b-it | 1.1% | 119,039 |
| gemma-2b | mlkro/gemma-2b-it-bnb-4bit-dolly-toxicity-study | 8.8% | N/A |
| gemma-2b | SongTonyLi/gemma-2b-it-SFT-D1 chosen-orca | 6.7% | 276 |
| gemma-2b | rootsec1/gemma-2B-it-customer-support | 2.4% | 64 |
| gemma-2-2b | google/gemma-2-2b | 6.6% | 330,898 |
| gemma-2-2b | google/gemma-2-2b-it | 0.6% | 364,325 |
| gemma-2-2b | mlkro/gemma-2-2b-it-bnb-4bit-dolly-toxicity-study | 6.0% | N/A |
| gemma-2-2b | IlyaGusev/gemma-2-2b-it-abliterated | 0.8% | 1,187 |
| gemma-2-2b | AXCXEPT/EZO-Common-T2-2B-gemma-2-it | 0.4% | 1,813 |

## B   Data & Code

The code used to conduct toxicity evaluations and fine-tune the models in this paper can be found at <code to be added following de-anonymization>.

The data used to fine-tune models was created by Databricks and can be accessed via Hugging Face at: https://huggingface.co/datasets/databricks/databricks-dolly-15k

