# OpenReview forum: "The effect of fine-tuning on language model toxicity"
_NeurIPS.cc/2024/Workshop/SafeGenAi — SafeGenAi Oral_

### Official Review · Reviewer_zaFD · 2024-10-08
**Review for The effect of fine-tuning on language model toxicity**

**Rating:** 8
**Confidence:** 4

**Review:**

This paper explores the impact of fine-tuning on the toxicity of large language models (LLMs), particularly when using parameter-efficient fine-tuning methods such as Low-Rank Adaptation (LoRA). It compares base models with their instruction-tuned variants, evaluates how fine-tuning with non-adversarial datasets affects toxicity, and investigates community-contributed model variants. The study includes models from Google, Meta, and Microsoft, assessing their toxicity rates across different prompts, including severe and non-toxic inputs. Overall, I think it is a great paper that makes a valuable contribution to understanding the effects of fine-tuning on LLM toxicity, particularly in the context of open models and community contributions
Some concerns:
(1)	The authors analyze multiple models, including two generations from major labs such as Google (Gemma), Meta (Llama), and Microsoft (Phi). This provides a clear picture and a robust set of empirical analysis. As author mentioned, there could be much more variations in terms of model size, etc (e.g. Llama 3.1-70B). I think authors can at least include the new Llama3.2 small family (1B & 3B) to verify the impacts of fine-tuning on model size (Gemma-2B). It would be better to also include potentially OpenAI’s close-source models.
(2)	I think author can also add adversarial testing to evaluate model robustness in the face of deliberate toxic inputs.
(3)	Lastly, I think author can also test the impact of LoRA parameters on final performance, as this is often of great practical importance to people doing fine-tuning; Given the size of model and finetuning dataset and the nature of the task, what would be the optimal LoRA parameter that you recommend?